# Regional Variations in Outpatient Antibiotic Prescribing in Germany: A Small Area Analysis Based on Claims Data

**DOI:** 10.3390/antibiotics11070836

**Published:** 2022-06-22

**Authors:** Oliver Scholle, Marieke Asendorf, Christoph Buck, Susann Grill, Christopher Jones, Bianca Kollhorst, Oliver Riedel, Benjamin Schüz, Ulrike Haug

**Affiliations:** 1Department of Clinical Epidemiology, Leibniz Institute for Prevention Research and Epidemiology—BIPS, Achterstrasse 30, 28359 Bremen, Germany; asendorf@leibniz-bips.de (M.A.); riedel@leibniz-bips.de (O.R.); haug@leibniz-bips.de (U.H.); 2Department of Biometry and Data Management, Leibniz Institute for Prevention Research and Epidemiology—BIPS, Achterstrasse 30, 28359 Bremen, Germany; buck@leibniz-bips.de (C.B.); susann.grill@gmail.com (S.G.); kollhorst@leibniz-bips.de (B.K.); 3Institute for Public Health and Nursing Research, University of Bremen, 28359 Bremen, Germany; jones@uni-bremen.de (C.J.); benjamin.schuez@uni-bremen.de (B.S.); 4Faculty of Human and Health Sciences, University of Bremen, 28359 Bremen, Germany

**Keywords:** antibiotics, drug utilization research, spatial analysis, regional variations

## Abstract

A comprehensive small area description of regional variations in outpatient antibiotic prescribing in Germany is lacking. Using the German Pharmacoepidemiological Research Database (GePaRD), a claims database covering ~20% of the German population, we determined the age- and sex-standardized prescription rates of antibiotics (number of outpatient prescriptions per 1000 persons/year). We calculated these prescription rates overall and on the level of 401 German districts for the calendar years 2010 and 2018. In 2018, the standardized prescription rate of antibiotics in the total study population was 23% lower than in 2010 (442 vs. 575 per 1000 persons/year). Among 0–17-year-olds, prescription rates across districts ranged from 312 to 1205 in 2010 and from 188 to 710 in 2018 per 1000 persons/year; among adults (≥18 years), they ranged from 388 to 841 in 2010 and from 300 to 693 in 2018 per 1000 persons/year. Despite the overall decline in outpatient antibiotic prescribing between 2010 and 2018, regional variations at the district level remained high in all age groups in Germany. Identifying reasons that explain the persistently high prescription rates in certain regions will be helpful in designing effective and tailored measures to further improve antibiotic stewardship in these regions.

## 1. Introduction

Antibiotics are important for the treatment of bacterial infections, and antibiotic resistance is one of the main challenges for health systems worldwide. Previous antibiotic exposure is a key driver of antibiotic resistance in humans [1]. Therefore, it is crucial to avoid inappropriate prescribing of antibiotics.

Compared to other European countries, Germany belongs among those with the lowest amount of outpatient antibiotics prescribed: in 2019, Germany ranked fifth among 30 countries reporting respective data to the European Surveillance System [2]. Nonetheless, there have been reports about pronounced regional variations across the 16 federal states in antibiotic prescribing, raising concerns about differences in the appropriateness of prescribing in Germany. An analysis based on data from 2018 including all persons with statutory health insurance (SHI) coverage—i.e., about 87% of the German population [3]—showed that the overall prescription rate of antibiotics is lower in federal states in eastern and southern Germany as compared with those in western and northern Germany [4]. In addition, it has been shown that regional prescription patterns for antibiotics differ by age group. Some federal states in eastern Germany (e.g., Saxony-Anhalt) had above-average prescription rates for children but below-average prescription rates for adults [4,5].

However, a small area (district-level) description of regional variations in outpatient antibiotic prescribing in Germany, i.e., beyond the federal states, is entirely lacking for adults and only available for the year 2010 for children and adolescents [6]. Such a description would, however, be helpful in identifying clear targets for interventions aiming to improve antibiotic stewardship. Understanding the reasons underlying these regional variations would also be essential to developing such interventions, but they are not well-studied. Consequently, the German Federal Ministry of Health commissioned a project called SARA (“Studie zur Analyse der Regionalen Unterschiede bei der Antibiotika-Verordnung” [Study to analyze regional variations in antibiotic prescriptions]), which is divided into two parts. In the first part, the project aims to describe the current status as well as changes in outpatient antibiotic prescribing, including a detailed description of regional variations using data from the German Pharmacoepidemiological Research Database (GePaRD). In the second part, reasons for regional variations in antibiotic prescribing will be explored based on a qualitative study (interviews with prescribers) and a quantitative study (multivariable analyses based on GePaRD).

In this paper, we present the study design and results of the first part of the SARA project, i.e., a description of the current status and changes in outpatient antibiotic prescribing in Germany in 2018 compared to 2010, with a special focus on regional variations in antibiotic prescribing on the level of 401 German districts (stratified by age groups).

## 2. Methods

### 2.1. Data Source

GePaRD is based on claims data from four SHI providers in Germany and includes information on persons who have been insured with one of the four participating SHI providers since 2004 or later [7,8]. Per data year, GePaRD covers approximately 20% of the general population, and all geographical regions of Germany are represented. Demographic data include sex, age, and region of residence on the district level (see details below). Prescription data in GePaRD include all drugs prescribed by general practitioners or specialists in the outpatient setting that SHI providers have reimbursed. Prescriptions are coded according to the German modification of the WHO Anatomical Therapeutic Chemical (ATC) classification system (version as of May 2019 for this study). 

### 2.2. Study Design, Study Population, and Measures of Interest

We conducted cross-sectional analyses for the calendar years 2010 and 2018. We selected the year 2010 to describe changes over a longer period of time and also to complement and facilitate comparisons with previous research which was restricted to children [6]. The year 2018 was selected because it was the most current data year available for analyses. We only included persons insured for at least one day in each quarter of the respective year in both calendar years. Further, we only included persons with information on sex (female or male) and birth year. For both calendar years, we determined the prescription rate as the primary measure of interest, defined as the total number of prescriptions per population in the respective year (in “prescriptions per 1000 persons/year”, i.e., the total number of antibiotic prescriptions/total number of persons in the study population). We also assessed the prescription prevalence, defined as the proportion of persons with at least one antibiotic prescription among all persons in the study population (in “%”).

### 2.3. Assessment of Antibiotic Prescriptions and Region of Residence

For each person, we assessed outpatient antibiotic prescriptions based on the ATC codes J01 (“antibacterials for systemic use”) and P01AB01 (“metronidazole”), using the date of prescription to assign them to calendar years. According to the classification in pharmacological subgroups (ATC 3rd level), individual antibiotic agents were grouped into subgroups (Appendix A).

For each person and separately for both calendar years, we determined the region of residence on the district level (“Kreise und kreisfreie Städte”). Since there have been changes in the classification of districts between 2010 and 2018—mainly because some districts were merged—we standardized all districts to the status of 2018. Accordingly, our analysis includes 401 districts.

### 2.4. Data Analysis

We calculated the prescription rate overall for both calendar years and stratified by sex and age (individual years and age groups adapted from earlier studies [4]). For 2018, we also determined the proportional distribution of antibiotic prescriptions by antibiotic subgroup in the same strata.

To describe regional variations in antibiotic prescribing at the district level and over time, we determined the prescription rate in each district for children below school age (≤6 years), for children and adolescents combined (0–17 years), and for adults (≥18 years) for both calendar years. For the calendar year 2018, we additionally calculated the prescription prevalence. We first excluded districts with fewer than 100 eligible insured persons in the corresponding age group for the graphical presentation to ensure reliable coverage. As a result, 4 of the 401 districts had to be excluded. To classify districts with lower or higher antibiotic prescribing, we then categorized the remaining districts—according to the prescription rate or the prescription prevalence, respectively—into five classes using the Jenks natural breaks classification method, which automatically identifies jumps in continuous data values and objectively maximizes the differences between classes [9]. The natural breaks classification method seeks to arrange values into discrete classes so that the difference between each value and the class mean is minimized and the difference between the means of different classes is maximized. We chose this method to handle skewness in the distribution of district-level prescription rates and prevalences. Because of the large differences in antibiotic prescribing between minors and adults and between calendar years, the estimated prescription rates and prevalences were categorized separately for each age group and calendar year.

To account for differences in the sex and age structure of the study population in the different regions and calendar years, prescription rates and prescription prevalences were directly standardized according to age/sex (as appropriate). The reference population for standardization was the German population as of 31 December 2017.

In this study, the term “eastern Germany” refers to the federal states Brandenburg, Mecklenburg-Western Pomerania, Saxony, Saxony-Anhalt, and Thuringia—i.e., the states of the former German Democratic Republic (“East Germany”)—as well as Berlin in its entirety (Appendix A).

Jenks natural breaks were calculated using R (version 3.6.0; Vienna, Austria), package BAMMtools (version 2.1.7) [10]. Descriptive analyses were conducted using SAS (version 9.4; SAS Institute, Cary, NC, USA). Maps were created using R, package sf (version 0.9-8) [11], or QGIS (version 3.24.1) [12]; spatial data on districts and federal states were used from the Federal Agency for Cartography and Geodesy (BKG).

## 3. Results

### 3.1. General Description of Prescription Rates

The entire study population comprised 13.4 and 16.7 million persons in 2010 and 2018, respectively (Table 1). More than 7 million outpatient antibiotic prescriptions were issued in both study years. From 2010 to 2018, the standardized prescription rate of antibiotics in the total study population declined by 23%, from 575 to 442 per 1000 persons/year. The relative decline decreased with age, from 52% in the age group 0–1 year to 15% in persons aged ≥ 65 years. This pattern did not substantially differ between females (Appendix A) and males (Appendix A). 

In both years, the standardized prescription rate of antibiotics for males and females combined was highest in the age group 2–5 years (2018: 568 per 1000 persons/year) and lowest in the age group 10–14 years (2018: 236 per 1000 persons/year) (Table 1). In the analyses stratified by sex, this age pattern was also seen for males in both years (Appendix A) and for females in 2010 (Appendix A), whereas, in females, prescription rates in 2018 ranked highest in those aged 2–5 years, 18–24 years, and ≥65 years (558–576 per 1000 persons/year). 

Appendix A shows prescription rates of antibiotics by age (in individual years) and sex in 2018. In both males and females, prescription rates peaked below the age of five years, reaching about 600 per 1000 persons/year and dropping sharply up to the age of 13. Beyond age 13, prescription rates in females peaked again at age 18, then remained between 480 and 600 per 1000 persons/year until age 80, and exceeded these levels afterward. In males, rates peaked at age 18, albeit at a lower level, and continuously increased from 25 years, reaching levels above 500 per 1000 persons/year from age 80 onwards. 

### 3.2. Proportional Distribution of Antibiotics by Antibiotic Subgroup 

Figure 1 shows the proportional distribution of antibiotic prescriptions in the total population by antibiotic subgroup in 2018. The share of basic penicillins and cephalosporins, respectively, was ~40–50% and ~30–40% in children up to 14 years, then decreased to levels of ~20–30% (basic penicillins) and ~15–20% (cephalosporins) up to the age of 54 years, and was below 20% for both antibiotic subgroups at older ages. Macrolides/lincosamides had a share of 19–24% from 10–14 to 55–64 years. The share of fluoroquinolones was ≥8% from age 18 years onwards and reached 20% among persons aged 65 years or older. As shown in Appendix A, the patterns were similar in females and males with some exceptions, such as a higher percentage of nitrofurantoin/fosfomycin/nitroxoline in females than in males (overall 12% vs. 1%). Appendix A provides detailed information on the distribution within antibiotic subgroups among both sexes, showing that prescribing basic penicillins was mainly limited to amoxicillin and phenoxymethylpenicillin, and, for cephalosporins, it was limited to second- and third-generation cephalosporins.

### 3.3. Regional Variations in Antibiotic Prescribing 

Figure 2 shows the standardized prescription rates of antibiotics among children and adolescents aged 0–17 years (n = 2,547,747) in the German districts in 2018 (thick lines depict the borders of federal states, labelled in Appendix A). Prescription rates ranged from 188 to 710 prescriptions per 1000 persons/year across districts. In the lowest class comprising 98 districts, prescription rates ranged from 188 to 313 per 1000 persons/year; in the highest class comprising 14 districts, prescription rates ranged from 560 to 710 per 1000 persons/year. High prescription rates were particularly found in districts near the western border of Germany (except for Baden-Württemberg), in Rhineland-Palatinate, and the northeast of Bavaria. The 14 districts with the highest prescription rates were all located in western Germany. In the subgroup of children aged ≤ 6 years (n = 983,049; Appendix A), the geographical pattern was the same as in all children and adolescents, but the prescription rates were higher, and the difference between the minimum and maximum was more pronounced (range from 249 to 1005 prescriptions per 1000 persons/year). In 2010, the standardized prescription rates among children and adolescents aged 0–17 years ranged from 312 to 1205 prescriptions per 1000 persons/year across districts; regarding geographical patterns, there were also districts in the class with the highest prescription rates in eastern Germany (Appendix A), which was no longer the case in 2018 as mentioned above. Results for the subgroup of children aged ≤ 6 years in 2010 are shown in Appendix A.

Figure 3 shows the standardized prescription rates of antibiotics among adults aged ≥ 18 years (n = 14,075,311) in the German districts in 2018. Prescription rates ranged from 300 to 693 per 1000 persons/year across districts. In the lowest class comprising 109 districts, prescription rates ranged from 300 to 410 per 1000 persons/year; in the highest class comprising 15 districts, prescription rates ranged from 598 to 693 per 1000 persons/year. The 15 districts with the highest prescription rates were near the western border of Germany (except for Baden-Württemberg) and in Rhineland-Palatinate. Almost all districts in eastern Germany had prescription rates in the two classes with the lowest prescription rates. In 2010, prescription rates among adults aged ≥ 18 years ranged from 388 to 841 prescriptions per 1000 persons/year across districts; the overall geographical pattern was similar to that in 2018 (Appendix A).

Regional variations in antibiotic prescribing based on the prescription *prevalence* in the different age groups are shown and described in the Appendix A.

## 4. Discussion

Our study on outpatient antibiotic prescribing in Germany found substantial regional variations in outpatient antibiotic prescribing at the district level in both minors and adults. In addition, the study showed a substantial decline in prescription rates: In 2018, we found an overall standardized prescription rate of 442 per 1000 persons/year, which was almost 25% lower compared to 2010. In 2018, prescription rates were above 500 per 1000 persons/year in the age groups 2–5 years and ≥65 years. However, compared to 2010, the decline was markedly higher in those aged 2–5 as compared to those ≥65 years (47% vs. 15%). Despite the overall decline in prescription rates, the relative differences in prescription rates between districts remained high in 2018—the rates differed up to almost fourfold among minors (0–17 years) and up to more than twofold among adults (≥18 years). In western Germany, prescription rates generally tended to be higher than in eastern Germany, and this difference increased in minors between 2010 and 2018. In all age groups, the highest prescription rates were found near the western border of Germany.

Our results regarding the overall pattern of outpatient antibiotic prescribing—i.e., independently of the region—are similar to those from Holstiege et al. [4]. This applies for antibiotic prescription rates by age group, the decline of prescription rates between 2010 and 2018, and the proportional distribution of antibiotics by antibiotic subgroup. Our study extends the results from Holstiege et al. [4] by a detailed description of regional variations on the district level with respect to changes over time and a sex-specific description of prescription patterns according to antibiotic subgroups. Several other studies on outpatient antibiotic prescribing in Germany have also investigated regional variations. However, these studies were based on older data [5,6,13] and/or did not analyze regional variations at the district level [4,5,13,14,15,16].

While the overall decline in prescription rates observed in our study suggests that the measures taken in the context of antibiotic stewardship programs in Germany were effective, our findings also illustrate a need for further improvement, particularly in certain regions. In 2018, there were still marked differences in prescription rates across districts unlikely to be explained by variations in the prevalence of infectious diseases. The differences may have been due to local cultural variation [17,18,19], e.g., the expectation of receiving antibiotic prescriptions. Regional factors related to the health system, such as the density of physicians, could also have played a role. The present study suggests that further activities of antibiotic stewardship should specifically focus on and be tailored to certain regions rather than applying uniform measures to all regions. Continuous monitoring of prescription rates on the district level—such as in our study—can provide useful information to assess whether this approach will lead to a further decline in prescription rates. Apart from overall prescription rates, our study also illustrates a need for improvement with respect to the subgroups of antibiotics prescribed. For example, cephalosporins accounted for a large share, particularly in children. This does not reflect German guideline recommendations, as already described and discussed by Holstiege et al. [14]. Further, the share of fluoroquinolones was relatively high, particularly among older persons, although they are considered reserve antibiotics, partly because of a higher risk profile than other antibiotics [20].

There are specific strengths and limitations to be considered in the interpretation of our results. Our study is the first to provide a small area description of regional variations in outpatient antibiotic prescribing on the district level in Germany for both minors and adults. The large differences in antibiotic prescribing between districts of the same federal state demonstrate that describing regional variations solely on the federal state level is not sufficient. The detailed small area description in our study was possible due to the large claims database used. The database is free of recall and non-responder bias, and it provides a valid denominator for calculations of prevalences or rates. It also includes outpatient data from all types of providers (general practitioners and specialists). Another strength of our study is that—in contrast to most earlier studies—we used more than one measure to evaluate regional variations in antibiotic prescribing (prescription rate and prescription prevalence). Following the rationale provided by Holstiege et al. [4], we used the prescription rate as the primary measure, mainly for the following reasons: (a) the frequency of prescribed antibiotics has been suggested to be more relevant for antibiotic resistance than the total amount of antibiotics prescribed based on the Defined Daily Doses (DDDs) [21]; (b) compared to measures based on the DDDs, the prescription rate facilitates comparisons of the frequency of prescribed antibiotics between different age groups regardless of age-related differences in dosages [22]; and (c) the prescription rate permits the assessment of time trends in antibiotic prescribing independent of changes in dosage over time [23,24].

Our study also has limitations. First, given that German claims data do not contain information on antibiotics prescribed in hospitals, our study focused on outpatient antibiotic prescribing. However, as the vast majority of antibiotics are prescribed in the outpatient sector [25], we think it is highly justified to put a special focus on this setting. Second, as usual in pharmacoepidemiological studies, there is uncertainty as to whether the dispensed antibiotics were actually taken. This also means that regional variations observed in our study may not always be identical to regional variations in the use of antibiotics. This would, for example, be the case if the strategy of “delayed prescribing”—i.e., a physician issues a prescription but advises the patient to use the antibiotic only if symptoms persist/worsen or once the test results are known—is not used uniformly across regions. Third, the regional variations we observed applied to the GePaRD population but may not necessarily be extrapolated to the whole German population. However, given the agreement of our results with Holstiege et al. [4] and the fact that their study was based on a full sample of persons covered by SHI in Germany, we conclude that GePaRD can be considered representative regarding antibiotic prescribing for all persons covered by SHI in Germany (~90% of the German population).

In conclusion, our study shows that—despite the overall decline in outpatient antibiotic prescribing between 2010 and 2018—regional variations on the district level remained high in all age groups in Germany. Identifying reasons that explain the persistently high prescription rates in certain regions will be helpful in designing effective and tailored measures to further improve antibiotic stewardship in these regions.

## Figures and Tables

**Figure 1 antibiotics-11-00836-f001:**
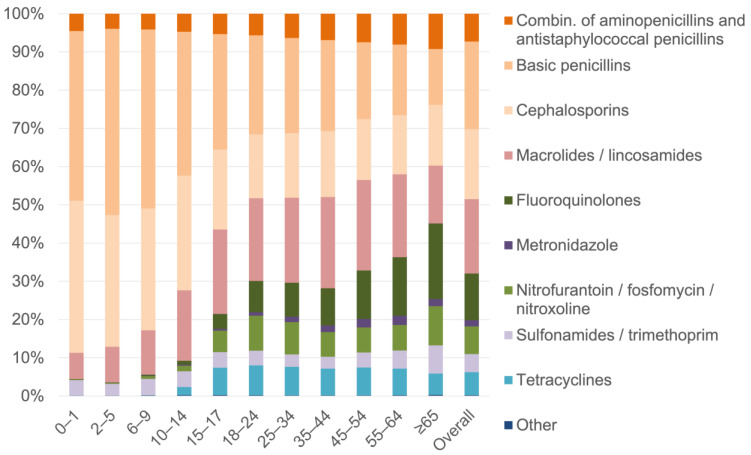
Proportional distribution of antibiotic prescriptions by antibiotic subgroup in 2018.

**Figure 2 antibiotics-11-00836-f002:**
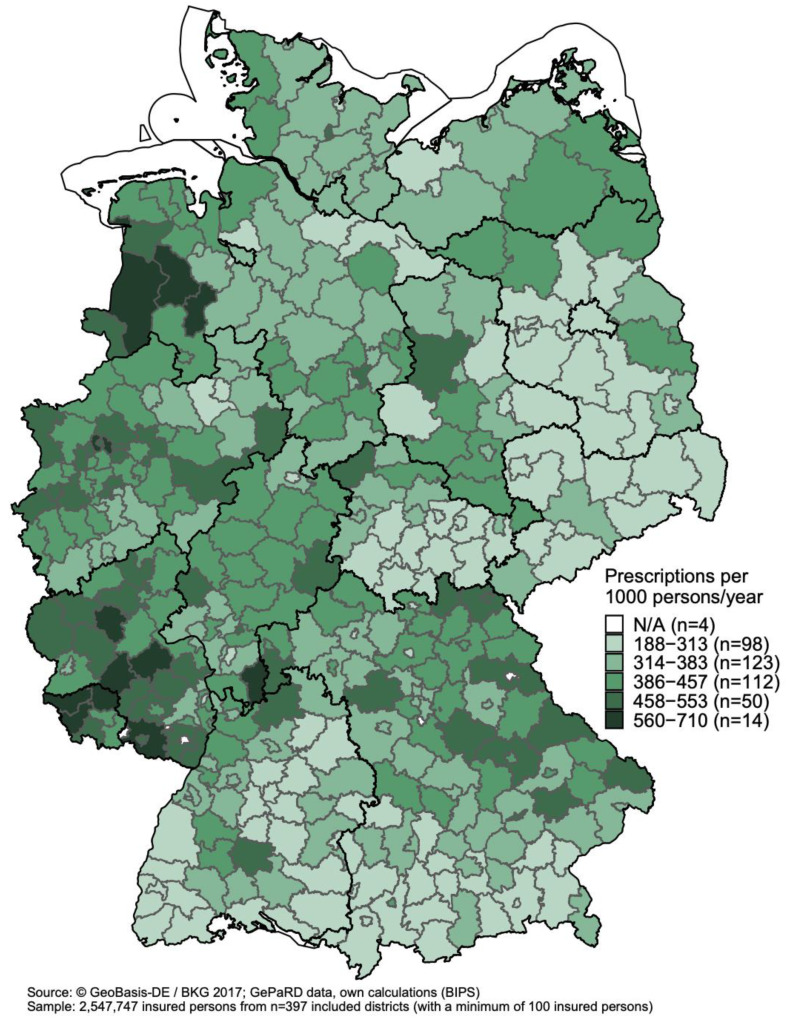
Age- and sex-standardized prescription rates (prescriptions per 1000 persons/year) of antibiotics among children and adolescents aged 0–17 years by district in 2018.

**Figure 3 antibiotics-11-00836-f003:**
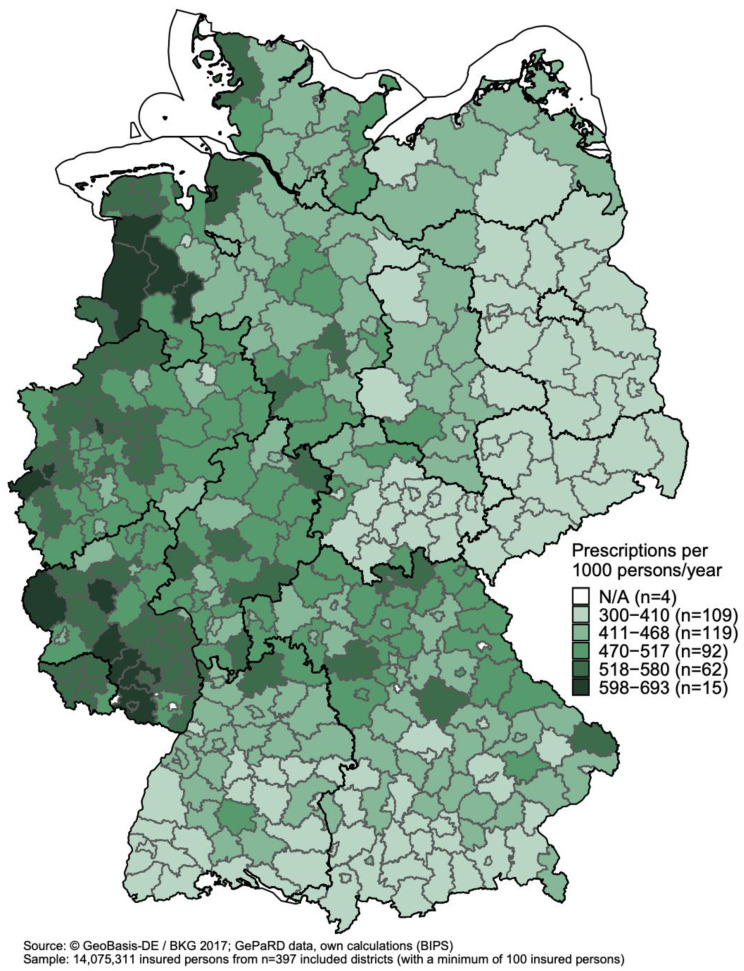
Age- and sex-standardized prescription rates (prescriptions per 1000 persons/year) of antibiotics among adults aged ≥ 18 years by district in 2018.

**Table 1 antibiotics-11-00836-t001:** Study population, antibiotic prescriptions, and standardized prescription rates of antibiotics in 2010 and 2018.

	Age Group in Years											Overall
	0–1	2–5	6–9	10–14	15–17	18–24	25–34	35–44	45–54	55–64	≥65	
Study population, n												
2010	136,559	436,886	480,760	701,457	378,154	1,013,358	1,688,664	1,837,739	2,293,396	1,734,896	2,685,105	13,386,974
2018	207,313	631,394	574,562	698,922	443,551	1,278,576	2,324,134	2,144,815	2,510,252	2,445,223	3,429,056	16,687,798
Antibiotic prescriptions, n												
2010	111,057	466,396	284,940	269,106	207,945	594,281	904,525	1,015,324	1,176,770	976,126	1,661,508	7,667,978
2018	82,842	359,160	211,225	165,442	158,872	538,842	920,133	956,370	1,077,739	1,161,307	1,829,845	7,461,777
Standardized prescription rates ^a^												
2010	651.3	1069.1	596.0	384.0	552.9	579.6	526.8	547.3	505.0	555.1	629.0	574.6
2018	313.7	568.4	366.4	236.4	357.8	420.0	391.5	441.4	421.8	469.0	533.6	442.0
Change in prescription rate from 2010 to 2018	−52%	−47%	−39%	−38%	−35%	−28%	−26%	−19%	−16%	−16%	−15%	−23%

^a^ Prescriptions per 1000 persons/year; age- and sex-standardized using the German population on 31 December 2017 as reference.

## Data Availability

As we are not the owners of the data, we are not legally entitled to grant access to the data of the German Pharmacoepidemiological Research Database. In accordance with German data protection regulations, access to the data is granted only to BIPS employees on the BIPS premises and in the context of approved research projects. Third parties may only access the data in cooperation with BIPS and after signing an agreement for guest researchers at BIPS.

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
