# Peer review of "Regional Variations in Outpatient Antibiotic Prescribing in Germany: A Small Area Analysis Based on Claims Data"

_antibiotics, 2022, doi:10.3390/antibiotics11070836_

Round 1
Reviewer 1 Report
Regional variations in outpatient antibiotic prescribing in Germany: a small area analysis based on claims data
Comments
The authors of this manuscript researched into antibiotic prescription in outpatients in various German regions. They found that the prescription rates were lower in a more recent year (2018) than an earlier year (2010) though some regions demonstrated high prescription rates. The manuscript is capable of impacting the field of antimicrobial resistance especially in the area of antibiotic stewardship. However, authors should address the following concerns observed.
Lines 13-16: This sentence “Using the German Pharmacoepidemiological Research Database (GePaRD), a claims database covering ~20% of the German population, we determined the age- and sex-standardized prescription rates of antibiotics (number of outpatient prescriptions per 1,000 persons/year)—overall and on the level of 401 German districts—for the calendar years 2010 and 2018” is too long make the information presented to be difficult to understand by readers. Please present it in smaller sentences.
Line 22: Replace “on” with “at”
Line 23: Replace “to design” with “in designing”
Lines 29-30: For better clarity, replace this sentence “A key driver of antibiotic resistance in humans is previous antibiotic exposure” with “Previous antibiotic exposure is a key driver of antibiotic resistance in humans”
Line 45: Add a comma after “states”
Line 47: Replace “to identify” with “in identifying”
Lines 48-50: For better presentation, authors should replace this sentence “To develop such interventions, knowledge on reasons underlying these regional variations would also be important, but they are not well studied either” with “Understanding the reasons underlying these regional variations would also be essential to developing such interventions, but they are not well studied.”
Line 54: Add a comma after “prescribing”
Line 71: Replace these words “….. have been reimbursed by SHI providers.” words with “……SHI providers have reimbursed.”
The authors stated the reason why 2018 was considered. Authors should also include the reasons why 2010 was surveyed among other years. Why did the authors not consider other years apart from 2010 and 2018?
Line 76: Replace these words “….at the time of analyses)” with “….for analyses)”
Lines 76-78: Rewrite this sentence “In both calendar years, we only included persons insured for at least one day in each quarter of the respective year.” for clarity as “We only included persons insured for at least one day in each quarter of the respective year in both calendar years.”
Line 81: Add “the” before “total”
Line 83: Please delete “which was”
Under the study design, authors should include the exclusion criteria developed in the selection of the participants.
How many participants did the authors select in both years? The sample size used should be stated in the method section.
Lines 97-98: Please rewrite this sentence “For both calendar years, we determined the prescription rate overall and stratified by sex and age (individual years and age groups, which were adapted from earlier studies” as “We determined the prescription rate overall for both calendar years and stratified by sex and age (individual years and age groups adapted from earlier studies”
Line 100: Add “an” before “antibiotics”
Lines 105-107: Rewrite this sentence “For the graphical presentation, we first excluded districts with fewer than 100 eligible persons in the corresponding age group to ensure reliable coverage. As a result, 4 of the 401 districts had to be excluded.” for clarify as “We first excluded districts with fewer than 100 eligible persons in the corresponding age group for the graphical presentation to ensure reliable coverage.”
Line 107: Please delete “into categories”
Line 113: For conciseness, please replace “in such a way” with “so”
Line 117: Delete “, they” for clarity
Line 127: Replace “used” with “using”
Lines 135-136: Rewrite this sentence “In both study years, more than 7 million outpatient antibiotic prescriptions were issued” for clarity as “More than 7 million outpatient antibiotic prescriptions were issued in both study years”
Line 138: Rewrite these words “…with age from 52% in age group…” as “…with age, from 52% in the age group…”
Line 147: Add “the” before “age”
Line 148: Add “the” before “age”
Lines 154-156: Rewrite this sentence “In both, males and females, prescription rates peaked below the age of five years reaching levels of about 600 per 1,000 persons/year and dropped sharply up to the age of 13 years” as “In both males and females, prescription rates peaked below the age of five years, reaching about 600 per 1,000 persons/year and dropping sharply up to the age of 13”
Line 157: Delete “at levels”
Line 158: Replace “afterwards” with “afterward”. Replace “as well” with a comma before “albeit”. Add a comma after “level”
Line 159: Delete “the age of” before “25” for conciseness
Line 167: For conciseness delete “age group”
Line 170: Add a comma after “exceptions”
Line 173: Delete “of” after “prescribing”
Line 181: Delete “which are” for conciseness
Line 187: Delete “in” before “the”
Line 203: Delete “prescriptions” after “693”
Line 220: Delete the comma after “both”
Line 224: Add a comma after “2010” and “and” before “the”
Line 233: Add “the” before “region”. Delete “true” after “holds”
Line 235: Add a comma after “2018”
Line 237: Delete “, also” before “with”, the comma after “time” and “by” before “a sex-specific”
Line 241: Change “on” to “at”
Line 247: Delete “regarding”
Line 251: Delete the comma before “rather”
Line 255: Replace “of” before “improvement” with “for”
Line 259: Replace “in part” with “partly”
Line 260: Replace “compared to” with “than” for clarity
Line 263: Delete the comma after “both”
Line 267: Add “and” before “it”
Line 278: Add “the” before “assessment”
Line 283: Add a comma after “[25]”
Line 299: Replace “to design” with “in designing”
Author Response
The authors of this manuscript researched into antibiotic prescription in outpatients in various German regions. They found that the prescription rates were lower in a more recent year (2018) than an earlier year (2010) though some regions demonstrated high prescription rates. The manuscript is capable of impacting the field of antimicrobial resistance especially in the area of antibiotic stewardship. However, authors should address the following concerns observed.
Lines 13-16: This sentence “Using the German Pharmacoepidemiological Research Database (GePaRD), a claims database covering ~20% of the German population, we determined the age- and sex-standardized prescription rates of antibiotics (number of outpatient prescriptions per 1,000 persons/year)—overall and on the level of 401 German districts—for the calendar years 2010 and 2018” is too long make the information presented to be difficult to understand by readers. Please present it in smaller sentences.
Authors’ response: Thank you for this suggestion. We have now divided the sentence into two sentences (line 13):
“Using the German Pharmacoepidemiological Research Database (GePaRD), a claims database covering ~20% of the German population, we determined the age- and sex-standardized prescription rates of antibiotics (number of outpatient prescriptions per 1,000 persons/year). We calculated these prescription rates overall and on the level of 401 German districts for the calendar years 2010 and 2018.”
Line 22: Replace “on” with “at”
Line 23: Replace “to design” with “in designing”
Lines 29-30: For better clarity, replace this sentence “A key driver of antibiotic resistance in humans is previous antibiotic exposure” with “Previous antibiotic exposure is a key driver of antibiotic resistance in humans”
Line 45: Add a comma after “states”
Line 47: Replace “to identify” with “in identifying”
Lines 48-50: For better presentation, authors should replace this sentence “To develop such interventions, knowledge on reasons underlying these regional variations would also be important, but they are not well studied either” with “Understanding the reasons underlying these regional variations would also be essential to developing such interventions, but they are not well studied.”
Line 54: Add a comma after “prescribing”
Line 71: Replace these words “….. have been reimbursed by SHI providers.” words with “……SHI providers have reimbursed.”
Authors’ response: Thank you for improving the language—we have implemented all the above corrections.
The authors stated the reason why 2018 was considered. Authors should also include the reasons why 2010 was surveyed among other years. Why did the authors not consider other years apart from 2010 and 2018?
Authors’ response: Thank you for pointing this out, indeed the explanation was missing. We chose 2010 because the only existing report on antibiotic consumption at the district level in Germany was from 2010 and restricted to children, i.e., there were no data from adults. By using the year 2010, we could thus fill this gap and at the same time, compare our findings for children to those from 2010 (i.e., another data source).
We have added the following (line 85):
“We selected the year 2010 to describe changes over a longer period of time and also to complement and facilitate comparison with previous research which was restricted to children [6]. The year 2018 was selected because it was the most current data year available for analyses.”
Regarding the question on why we did not consider other years in between in addition to the years 2010 and 2018: We wanted to look at changes in regional variations over a longer period of time. Analyzing annual trends—particularly on the district level—would have made the amount of data to be conveyed to the reader much larger. This, in our view, would have been outside the scope of this paper.
Line 76: Replace these words “….at the time of analyses)” with “….for analyses)”
Lines 76-78: Rewrite this sentence “In both calendar years, we only included persons insured for at least one day in each quarter of the respective year.” for clarity as “We only included persons insured for at least one day in each quarter of the respective year in both calendar years.”
Line 81: Add “the” before “total”
Line 83: Please delete “which was”
Authors’ response: Thank you for these corrections—we have implemented all of them.
Under the study design, authors should include the exclusion criteria developed in the selection of the participants.
Authors’ response: Apart from the inclusion criteria (Methods; line 89), we did not develop any further criteria, i.e., there were no exclusion criteria. For the presentation in maps, we excluded districts with fewer than 100 eligible insured persons in the corresponding age group (see line 130), so this was not an exclusion criterion applied to the study population but only concerned the presentation on the district level.
How many participants did the authors select in both years? The sample size used should be stated in the method section.
Authors’ response: We agree that it is important to report on sample size. Given that sample size is a result of applying inclusion criteria, i.e., requires data analysis, we hope it is acceptable that we report the sample size at the beginning of the results section rather than in the methods section (see line 161).
Lines 97-98: Please rewrite this sentence “For both calendar years, we determined the prescription rate overall and stratified by sex and age (individual years and age groups, which were adapted from earlier studies” as “We determined the prescription rate overall for both calendar years and stratified by sex and age (individual years and age groups adapted from earlier studies”
Authors’ response: Thank you—we have implemented this correction and changed the sentence structure accordingly. As the next sentence also used the verb “determined” we changed the verb to “calculated” here (line 123).
Line 100: Add “an” before “antibiotics”
Authors’ response: That would have changed the meaning of the sentence; to make it clearer, the sentence now reads as follows (line 124):
“For 2018, we also determined the proportional distribution of antibiotic prescriptions by antibiotic subgroup in the same strata.”
Lines 105-107: Rewrite this sentence “For the graphical presentation, we first excluded districts with fewer than 100 eligible persons in the corresponding age group to ensure reliable coverage. As a result, 4 of the 401 districts had to be excluded.” for clarify as “We first excluded districts with fewer than 100 eligible persons in the corresponding age group for the graphical presentation to ensure reliable coverage.”
Line 107: Please delete “into categories”
Line 113: For conciseness, please replace “in such a way” with “so”
Authors’ response: Thank you—we have implemented all the above corrections.
Line 117: Delete “, they” for clarity
Authors’ response: That would have changed the meaning of the sentence; to make it clearer, the sentence now reads as follows (line 141):
“Because of the large differences in antibiotic prescribing between minors and adults and between calendar years, the estimated prescription rates and prevalences were categorized separately for each age group and calendar year.”
Line 127: Replace “used” with “using”
Lines 135-136: Rewrite this sentence “In both study years, more than 7 million outpatient antibiotic prescriptions were issued” for clarity as “More than 7 million outpatient antibiotic prescriptions were issued in both study years”
Line 138: Rewrite these words “…with age from 52% in age group…” as “…with age, from 52% in the age group…”
Line 147: Add “the” before “age”
Line 148: Add “the” before “age”
Lines 154-156: Rewrite this sentence “In both, males and females, prescription rates peaked below the age of five years reaching levels of about 600 per 1,000 persons/year and dropped sharply up to the age of 13 years” as “In both males and females, prescription rates peaked below the age of five years, reaching about 600 per 1,000 persons/year and dropping sharply up to the age of 13”
Line 157: Delete “at levels”
Line 158: Replace “afterwards” with “afterward”. Replace “as well” with a comma before “albeit”. Add a comma after “level”
Line 159: Delete “the age of” before “25” for conciseness
Line 167: For conciseness delete “age group”
Line 170: Add a comma after “exceptions”
Line 173: Delete “of” after “prescribing”
Line 181: Delete “which are” for conciseness
Line 187: Delete “in” before “the”
Line 203: Delete “prescriptions” after “693”
Line 220: Delete the comma after “both”
Authors’ response: Thank you for the above corrections, which we all implemented.
Line 224: Add a comma after “2010” and “and” before “the”
Authors’ response: That would have changed the meaning of the sentence; to make it clearer, this part now reads as follows (line 282):
“In 2018, prescription rates were above 500 per 1,000 persons/year in the age groups 2–5 years and ≥65 years. However, compared to 2010, the decline was markedly higher in those aged 2–5 as compared to those ≥65 years (47% vs. 15%).”
Line 233: Add “the” before “region”. Delete “true” after “holds”
Authors’ response: Thank you for pointing this out. We hope the reviewer agrees that we preferred “applies” instead of “holds” (line 292):
“This applies for antibiotic prescription rates by age group, the decline of prescription rates between 2010 and 2018, as well as the proportional distribution of antibiotics by antibiotic subgroup.”
Line 235: Add a comma after “2018”
Line 237: Delete “, also” before “with”, the comma after “time” and “by” before “a sex-specific”
Line 241: Change “on” to “at”
Line 247: Delete “regarding”
Line 251: Delete the comma before “rather”
Line 255: Replace “of” before “improvement” with “for”
Line 259: Replace “in part” with “partly”
Line 260: Replace “compared to” with “than” for clarity
Line 263: Delete the comma after “both”
Line 267: Add “and” before “it”
Line 278: Add “the” before “assessment”
Line 283: Add a comma after “[25]”
Line 299: Replace “to design” with “in designing”
Authors’ response: Thank you for these corrections, which we all implemented.
Reviewer 2 Report
Overall an interesting, straightforward and well-written study. I believe the article is suitable for publication in the journal. I really only have minor suggestions. Authors can correct the comments below and there is no further need for this reviewer to check the revised paper.
Lines 37 and 42. I would suggest a point "." instead of ":" at "...Germany: An analysis..." and "...group: Some federal..."
Line 45: "...Germany, i.e., beyond the federal states" I'd suggest adding a comma: "...Germany, i.e., beyond the federal states,"
Author Response
Overall an interesting, straightforward and well-written study. I believe the article is suitable for publication in the journal. I really only have minor suggestions. Authors can correct the comments below and there is no further need for this reviewer to check the revised paper.
Lines 37 and 42. I would suggest a point "." instead of ":" at "...Germany: An analysis..." and "...group: Some federal..."
Line 45: "...Germany, i.e., beyond the federal states" I'd suggest adding a comma: "...Germany, i.e., beyond the federal states,"
Authors’ response: Thank you for the positive remarks and for improving the language—we have implemented all corrections.